

# China's Clean Air Action has suppressed unfavorable influences of climate on wintertime PM$_{2.5}$ concentrations in Beijing since 2002

Meng Gao[1,2], Zirui Liu[3], Bo Zheng[4,5], Dongsheng Ji[3], Peter Sherman[6], Shaojie Song[2], Jinyuan Xin[3], Cheng Liu[7], Yuesi Wang[3], Qiang Zhang[4], Zifa Wang[3], Gregory R. Carmichael[8], Michael B. McElroy[2,6]

1 Department of Geography, Hong Kong Baptist University, Hong Kong SAR, China
2 John A. Paulson School of Engineering and Applied Sciences, Harvard University, Cambridge, MA, USA
3 State Key Laboratory of Atmospheric Boundary Layer Physics and Atmospheric Chemistry, Institute of Atmospheric Physics, Chinese Academy of Sciences, Beijing, China
4 Ministry of Education Key Laboratory for Earth System Modeling, Department for Earth System Science, Tsinghua University, Beijing, China
5 Laboratoire des Sciences du Climat et de l'Environnement, CEA-CNRS-UVSQ, UMR8212, Gif-sur-Yvette, France
6 Department of Earth and Planetary Sciences, Harvard University, Cambridge, Massachusetts 02138, United States
7 University of Science and Technology of China, Hefei, Anhui, China
8 Department of Chemical and Biochemical Engineering, University of Iowa, Iowa City, IA, USA

Correspondence: Meng Gao (mmgao2@hkbu.edu.hk), Zifa Wang (zifawang@mail.iap.ac.cn) and Michael B. McElroy (mbm@seas.harvard.edu)



## Abstract

Severe wintertime $PM_{2.5}$ pollution in Beijing has been receiving increasing worldwide attention, yet the decadal variations remain relatively unexplored. Combining field measurements and model simulations, we quantified the relative influences of anthropogenic emissions and meteorological conditions on $PM_{2.5}$ concentrations in Beijing over winters of 2002-2016. Between the winters of 2011 and 2016, stringent emission control measures resulted in a 21% decrease in mean mass concentrations of $PM_{2.5}$ in Beijing, with 7 fewer haze days per winter on average. With fixed emissions, meteorological conditions over the study period would have led to an increase of haze in Beijing, but the strict emission control measures have suppressed the unfavorable influences of recent climate. The unfavorable meteorological conditions are attributed to the weakening of the East Asia Winter Monsoon associated particularly with an increase in pressure associated with the Aleutian low.



## 1 Introduction

In recent years, persistent and severe haze episodes with high $PM_{2.5}$ concentrations occur frequently in China, attracting worldwide attention (*Cheng et al., 2016; Gao et al., 2016*). High aerosol concentrations during haze have been reported to cause traffic jams and flight cancelations (*Wu et al., 2005*), and have been linked to health damages (*Dockery et al., 1993*), climate change (*Ramanathan and Carmichael, 2008*) and ecosystem degradation (*Chameides et al., 1999*). The annual mean $PM_{2.5}$ concentrations in Beijing exceeded 90 µg/m$^3$ in 2013, nearly twice the China's National Ambient Air Quality Standard (NAAQS) of 35 µg/m$^3$ (*MEP, 2012*). January 2013 was reported as the haziest month over the past 60 years in Beijing, with maximum hourly and daily mean $PM_{2.5}$ concentrations exceeding 1000 µg/m$^3$ and 500 µg/m$^3$, respectively (*Wang et al., 2014*).

Since then, the State Council of China issued the Air Pollution Prevention and Control Action Plan (APPCAP, denoted as the Clean Air Action hereafter), which describes explicitly the pollution control measures, and proposed specific goals for concentrations by 2017 (*China State Council, 2013*). This action has been considered the most stringent air pollution control policy in Chinese history. The Jing-Jin-Ji, Yangtze River Delta, and Peral River Delta regions were required to reduce annual mean $PM_{2.5}$ concentrations by 15-25% compared with the concentrations in 2013, and the annual mean concentrations of $PM_{2.5}$ in Beijing should not exceed 60 µg/m$^3$ (*Cheng et al., 2018; China State Council, 2013*). Specific control measures included eliminating small coal-fired boilers, phasing out small, high-emitting factories, installing control facilities for emissions of VOCs (volatile organic compounds), and replacing residential coal burning with electricity and natural gas among others. (*Zheng et al., 2018*). With these rigorous control measures, China has made impressive progress, with annual mean $PM_{2.5}$ concentrations reduced in major metropolitan regions by 28-40% between 2013 and 2017 (*Zheng et al., 2018*).

A number of studies have used visibility as a surrogate to indicate trends of haze pollution in China over the past several decades (*Che et al., 2009; Chen and Wang, 2015; Ding and Liu, 2014; Wang and Chen, 2016*). *Chen and Wang (2015)* reported that haze days increased rapidly in the 1970s and remained relatively stable up to present. However, *Che et al. (2009)* illustrated





that there was a decreasing haze trend in winter for many cities over the interval 1981-2005.

*Wang et al. (2019)* argued that visibility is impacted significantly by meteorological factors, especially relative humidity, and thus visibility does not accordingly reflect the real changes in air pollution. Due to the lack of long-term measurements of $PM_{2.5}$ in China, limited studies have been conducted exploring the roles of anthropogenic emissions and meteorological conditions for the long-term variations of $PM_{2.5}$ in China. *Yang et al. (2016)* used the GEOS-

Chem model to simulate $PM_{2.5}$ in China between 1985 and 2005, and concluded that the increase of winter $PM_{2.5}$ was dominated over this period by the increase in anthropogenic emissions. They found that weakening of winds was the dominant meteorological factor. While this study has explored the relative roles of emissions and meteorology, no model validation of $PM_{2.5}$ was provided, and the recent decades were not covered in the study period. Long-term

measurements of $PM_{2.5}$ in Beijing reveal a slight decreasing trend of annual mean concentration over 2004-2012 (*Liu et al., 2015*). With the increase in availability of recent measurements, a closer reading of long-term variations is needed to better define the relative roles of anthropogenic emissions and meteorology.

In this study, we present a comprehensive analysis of the decadal trend of wintertime $PM_{2.5}$ in

Beijing based on regional meteorology-chemistry modeling, a new decadal emission inventory, and long-term observations of $PM_{2.5}$, including their composition. We address the following questions: (1) the influences of decadal changes in anthropogenic emissions and meteorology on the variations of winter $PM_{2.5}$ in Beijing; and (2) the key driving factors for the decadal variation of meteorology. The descriptions of model, emissions, and numerical experiments are

presented in Sect. 2. The two questions highlighted above are addressed in detail in Sect. 3; Sect. 4 provides an overall summary of the study.

## 2 Model Description and Configurations

### 2.1 Meteorology-chemistry modeling

We used the WRF-Chem (Weather Research and Forecasting model coupled with chemistry) model version 3.6.1 to simulate meteorology and emissions, transport, mixing, and the chemical transformation of trace gases and aerosols. Two nested domains were applied with





the outer domain covering East Asia and part of Southeast Asia, with the inner domain focusing

on North China (Fig. 1). Horizontal resolutions of 81km and 27km were configured

respectively for these two domains, and the model accounted for 27 vertical layers extending

from the surface to 50 hPa. The gas phase chemical mechanism CBMZ (*Zaveri and Peters,*

*1999*) coupled with the 8-bin sectional MOSAIC model with aqueous chemistry (*Zaveri et al.,*

*2008*) was adopted. The model treats all the important aerosol species, including sulfate, nitrate,

chloride, ammonium, sodium, black carbon (BC), primary organic and inorganic material. The

Fast-j radiation scheme (*Wild et al., 2000*) was selected to calculate photolysis rates. These

configurations have been shown in previous studies (*Gao et al., 2016a, 2016b, 2017*) to be

capable of reproducing winter haze episodes in North China.

## 2.2 Emissions

The monthly Multi-resolution Emission Inventory for China (MEIC, http://www.meicmodel.

org/) covering the years 2002-2017 (*Zheng et al., 2018*) was used for anthropogenic emissions.

This inventory considers emissions of sulfur dioxide ($SO_2$), nitrogen oxides ($NO_x$), carbon

monoxide (CO), non-methane volatile organic compounds (NMVOC), ammonia ($NH_3$), black

carbon (BC), organic carbon (OC), $PM_{2.5}$, $PM_{10}$, and carbon dioxide ($CO_2$) associated with

power generation, and the industrial, residential, transportation, and agricultural sectors. The

trends of wintertime emissions of these species in Beijing-Tianjin-Hebei region over the

interval 2002-2016 are displayed in Fig. S1. Emissions of $SO_2$ have decreased continuously

since 2004, while $NO_x$ emissions have declined since 2011. Emissions of all involved species

have decreased rapidly since 2012. Biogenic emissions were calculated online using the

MEGAN model (*Guenther et al., 2006*). Emissions from biomass burning were taken from the

GFED v3 dataset (*Randerson et al., 2015*).

## 2.3 Numerical experiments

Meteorological initial and boundary conditions were obtained from the National Centers for

Environmental Prediction (NCEP) Final Analysis (FNL) dataset. Chemical initial and

boundary conditions were taken from the climatological data provided by the NOAA

Aeronomy Lab Regional Oxidant Model (NALROM). Wintertime periods defined as the last





month of the year and the following two months of the next year were simulated for years 2002-2016. For example, the winter of 2002 includes December of 2002, January of 2003 and February of 2003. Nudging (assimilation) of the analyses was applied to produce realistic meteorological simulations. The simulations were conducted month by month (15 years × 3 month/year = 45 months). To overcome the impacts of initial conditions, five more days were simulated for each month and discarded as spin-up. Two sets of simulations were performed to elucidate the relative roles of changes in anthropogenic emissions and meteorological conditions: (1) CTL simulation, simulations of winter periods from 2002 to 2016 with varying meteorological conditions and anthropogenic emissions; and (2) MET simulation, simulations of winter periods from 2002 to 2016 with varying meteorological conditions only, with anthropogenic emissions fixed at levels that applied in 2002.

## 3 Results and Discussion

### 3.1 Model evaluation

Model evaluation was conducted in terms of both $PM_{2.5}$ concentrations and $PM_{2.5}$ chemical compositions, using measurements from urban Beijing (location is marked with red dot in Fig. 1). $PM_{2.5}$ was measured using the Two tapered Element Oscillating Microbalances (TEOM) system at the Institute of Atmospheric Physics site. More descriptions of the observations were archived in *Liu et al. (2015)*. Fig. 2 displays the variations of simulated and observed daily mean $PM_{2.5}$ concentrations over winters from 2002 to 2016. Temporal variations of simulated and observed $PM_{2.5}$ concentrations are generally consistent, with correlation coefficients ranging from 0.75 to 0.83 (Fig. 2). Notably, the model overestimates $PM_{2.5}$ concentrations for all periods. However, the overestimations decline gradually over time. Values of the mean bias decrease from 54.1 $\mu g/m^3$ over the 2004-2006 period to 15.9 $\mu g/m^3$ for recent winters. The broad ranges of errors in different periods reflect the changing uncertainty of emission inventories for different periods. In early times, documentations of emission sources were not as comprehensive as those available for recent years, leading to larger errors in early inventories.

According to results reported by *Zheng et al. (2018)*, $SO_2$ emissions decreased by 59%, $NO_x$





emissions decreased by 21%, BC emissions decreased by 28%, and OC emissions decreased
by 32% for China between 2013 and 2017. These remarkable changes in emissions are
expected to lead to notable changes in both the abundance and composition of $PM_{2.5}$. As shown
in Fig. 3, sulfate and OC exhibit the largest declines, ammonium and BC show slight decreases,
while nitrate concentrations remain relatively stable over 2013-2017. These measured trends
are captured generally well by the model, except that sulfate is still underestimated, and BC is
overestimated in Beijing. The underestimate of sulfate by models and the overestimate of BC
in Beijing have been well documented in previous studies (*Cheng et al., 206; Gao et al., 2016a,*
*2018a; Song et al., 2018*), attributed to missing reaction pathways and aging/deposition
treatments in models (*Song et al., 2019*). Several heterogeneous reaction pathways for sulfate
formation have been proposed, including the oxidation of $SO_2$ by $NO_2$, transition-meta-
catalyzed $O_2$, or $H_2O_2$ in aerosol water, and by $NO_2$ or $O_2$ on aerosol surfaces (*Cheng et al.,*
*2016; He et al., 2014; Hung et al., 2018; Li et al., 2018*). More recently, *Song et al. (2019)*
proposed that the heterogeneous production of hydroxymethanesulfonate (HMS) from the
reaction of $SO_2$ and formaldehyde could be an important chemical mechanism for wintertime
haze in China. With the rapid declines in sulfate in Beijing, the relative importance of nitrate
in $PM_{2.5}$ is enhanced, a circumstance worthy of special attention for future pollution control
policy.


**3.2 Influences of Anthropogenic Emissions and Meteorological Conditions on Haze in
Beijing**

Fig. 4 illustrates the wintertime mean concentrations of $PM_{2.5}$ and the numbers of haze days
from the CTL and MET simulations. Haze days are defined as occasions with daily mean
concentrations exceeding 150 $\mu g/m^3$. With fixed anthropogenic emissions, wintertime
averaged concentrations of $PM_{2.5}$ would have increased at a rate of 2.1 $\mu g/m^3$/year in Beijing
(Fig. 4a). Due to the implementation of China's Clean Air Action, the wintertime averaged
concentrations of $PM_{2.5}$ have been declining since 2012. Over 2002-2016 winters, mean
concentrations of $PM_{2.5}$ in Beijing decreased at a rate of 1.4 $\mu g/m^3$/year. Compared to
concentrations in the MET simulation, the mean mass concentrations of $PM_{2.5}$ decreased by 21%
in Beijing over the winters of 2011-2016.



The effectiveness of China's Clean Air Action has been highlighted also in the decline in the number of haze days in Beijing. In the MET simulation, the total number of haze days over 2011-2016 winters amounted to 157 days, reduced by 44 days as a result of the emission
controls implemented over this period (Fig. 4b). On average, China's Clean Air Action resulted in 7 fewer haze days per winter over 2011-2016. Over the entire study period, China's Clean Air Action altered the direction of changes in wintertime haze days, with rates changing from 0.8 day/year to -0.3 day/year. The differences in both mean concentrations of $PM_{2.5}$ and numbers of haze days from the two sets of simulations underscore the impressive success of
China's Clean Air Action. Given the more serious overestimate of $PM_{2.5}$ in early periods (Fig. 2), the declining rates inferred from the CTL simulation might have been slightly overstated. With unchanged anthropogenic emissions, the increasing trend of haze pollution that would have occurred for Beijing highlights the unfavorable influences of recent changes in local meteorology.

Using similar approach, *Cheng et al. (2019)* found that meteorological conditions explain 12.1% of the improved $PM_{2.5}$ air quality during 2013-2017, while large portions of the improvement are dominated by local (65.4%) and regional (22.5%) emission reductions. The current study examines longer term trend since 2002, but the more favorable meteorological conditions mentioned in *Cheng et al. (2019)* are illustrated also in Fig. 4. Our findings highlight also the
significance of emission reductions, especially after 2013, while the long-term trend of meteorological conditions since 2002 differs from it during 2013-2017.

### 3.3 Significance of Different Meteorological Variables

To identify the key meteorological variables for the unfavorable influences on air quality, we
applied the stepwise linear regression model (SLR) to determine the relative significance of multiple meteorological variables in terms of their contributions to the variations of $PM_{2.5}$ concentrations. In a SLR model, the selected predictors are included in the regression equation one by one. The predictor that contributes the most to the model is included first, and the process is continued if the additional predictor can statistically improve the regression (*Bendel*
*and Afifi, 1977*). Thus, the SLR model is widely used to select meaningful predictors. In this study, boundary layer heights (BLH), precipitation (PREC), near surface relative humidity





(RH2), near surface temperature (T2) and near surface wind speeds (WS10) were selected as predictors for the SLR model. These variables were extracted from the WRF-Chem meteorological simulations with analyses nudging applied. Table 1 summarizes the p values for each predictor. RH2 and WS10 were selected as the most significant predictors for wintertime $PM_{2.5}$ in Beijing (p values < 0.05). The influence of WS10 is greater than that for RH2. *Shen et al. (2018)* reported also that RH and meridional wind speeds drive stagnation and chemical production of $PM_{2.5}$ in Beijing.

The impacts of RH on aerosol composition and processes in winter were examined using measurements in Beijing, and the largest impacts were found for the growth of sulfate and organic aerosols associated with coal combustion (*Sun et al., 2013; Wang et al., 2017*). Although RH has been shown to be a good predictor for $PM_{2.5}$, it decreases slightly (-5.3 %/decade) between 2002 and 2016 period (Fig. 5a), which contradicts the predicted increasing trend of $PM_{2.5}$ under fixed emissions (Fig. 4a). Thus, the variability of RH2 is unlikely to be the driver of enhanced $PM_{2.5}$ under changing conditions of climate.

Between 2002 and 2016, simulated wintertime WS10 in Beijing declined gradually at a rate of 0.3 m/s/decade (Fig. 5b), in agreement with the declining trends inferred from observations (Fig. S2). Weaker wintertime near surface wind speeds are associated with enhanced $PM_{2.5}$ concentrations and increasing numbers of haze days since 2002. In winter, the North China plain features northwesterly winds associated with the East Asian Winter Monsoon (EAWM), properties of which depend largely on the development of both the Siberian high and the Aleutian low (*Jhun and Lee, 2004*). We calculated the intensity of the EAWM index using the pressure difference between area mean sea level pressure (SLP, data were taken from MERRA-2 reanalysis) over the region 90º E-110º E, 40º N-50º N and the area mean SLP over the region 120º -170º E, 40º-60º N. These two regions represent the central focal areas of the Siberian high and the Aleutian low, respectively. Fig. 5(c) indicates that the variations of EAWM intensity has declined gradually over time, consistent with the variations of wind speeds in Beijing. The weakening of EAWM intensity is due partially to the increasing pressure in the regions of the Aleutian low. As shown in Fig. 6, there is no significant change in SLP over the regions of the Siberian high, but SLP of the Aleutian low intensity decreased significantly between 2002 and 2016 (more than 50 Pa/year). Although no significant trend was observed





for the strength of the Siberian high, the changes in the position of the Siberian high has been linked to wintertime air quality in China over the past decades (*Jia et al., 2015*).

*Yin et al. (2015)* found a significant negative correlation between winter haze and the East Asia

Winter Monsoon from 1986 to 2010. Deterioration of air quality in China has been linked also to a weaker East Asia Summer Monsoon (*Chin et al., 2012*). The slacking of winds is observed not only for China (*Sherman et al., 2017*), but also for other countries including India (*Gao et al., 2018b*). Analyses using climate projections suggest that wind speeds in continental regions in the Northern Hemisphere will continue to decline under a warming climate (*Karnauskas et*

*al., 2018*), imposing greater pressure on measures for future control of air pollution.

## 4 Summary

Combining field measurements and model simulations, we quantified the relative influences on $PM_{2.5}$ concentrations in Beijing of anthropogenic emissions and meteorological conditions

over winters of 2002-2016. China's Clean Air Action has been effective in reducing both mass concentrations of $PM_{2.5}$ and the number of haze days. With fixed emissions, meteorological conditions over the study period should have resulted in an increase in haze in Beijing, but the strict emission control measures implemented by the government have suppressed the unfavorable influences associated with recent climate. Using a statistical method, we concluded

that RH2 and WS10 offer useful predictors for wintertime $PM_{2.5}$ in Beijing, with the variations of WS10 in particularly good agreement with the increasing trend of $PM_{2.5}$ concentrations with fixed emissions. The increasing trend of $PM_{2.5}$ under unfavorable meteorological conditions was attributed further to the weakening of Aleutian low and the EAWM. The variations of $PM_{2.5}$ concentrations were investigated in this study for the urban Beijing region, so were the

changes in wind speeds. We do not exclude the possibility that the trend may have been influenced by other factors, including for example increases in surface roughness (*Vautard et al, 2010*).




**4 figures are listed in the supplement.**

**Author contribution**

M.G. and M.B.M designed the study; M.G. performed model simulations and analyzed the data

with the help from S.S., P.S., and G.R.C.; B. Zheng and Q. Zhang provided the emission

inventory; Z.W., Y.W., Z.L., D.J., J.X., and C.L. provided measurements. M.G. and M.B.M.

wrote the paper with inputs from all other authors.

**Data availability**

The measurements and model simulations data can be accessed through contacting the

corresponding authors.

**Competing interests**

The authors declare that they have no conflict of interests.


**Acknowledgement**

This work is supported by the Harvard Global Institute.

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





Table 1. p-values for stepwise linear regression model for Beijing.

| Meteorological Variables | p-values |
| --- | --- |
| BLH | 0.38 |
| PREC | 0.64 |
| RH2 | 0.02 |
| T2 | 0.95 |
| WS10 | 0.00 |

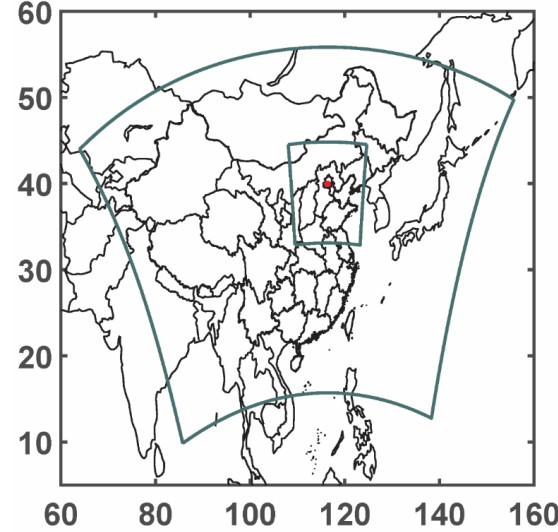

460       Fig.1. WRF-Chem modeling domain settings and locations of observations.



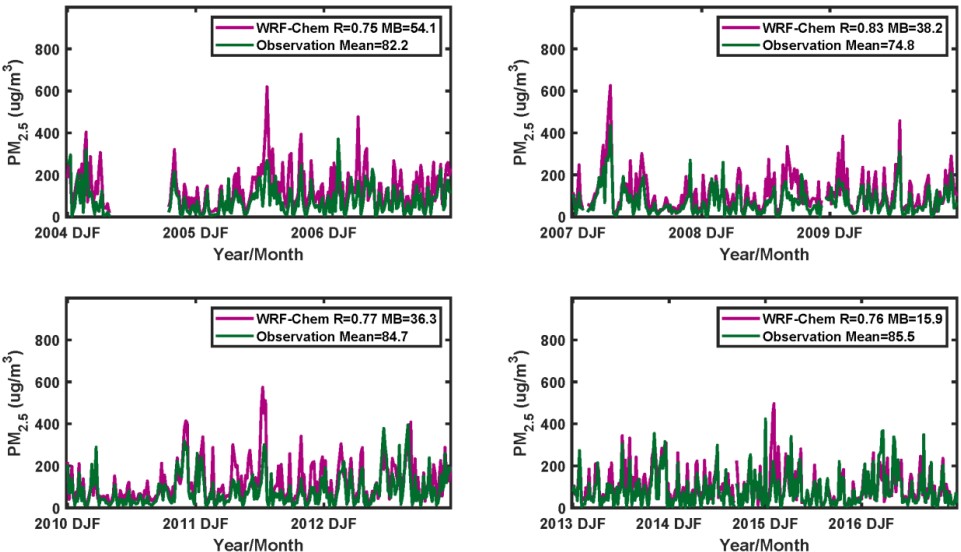

Fig. 2. Simulated and observed temporal variations of daily mean PM$_{2.5}$ concentrations in Beijing,
with correlation coefficient and mean bias.

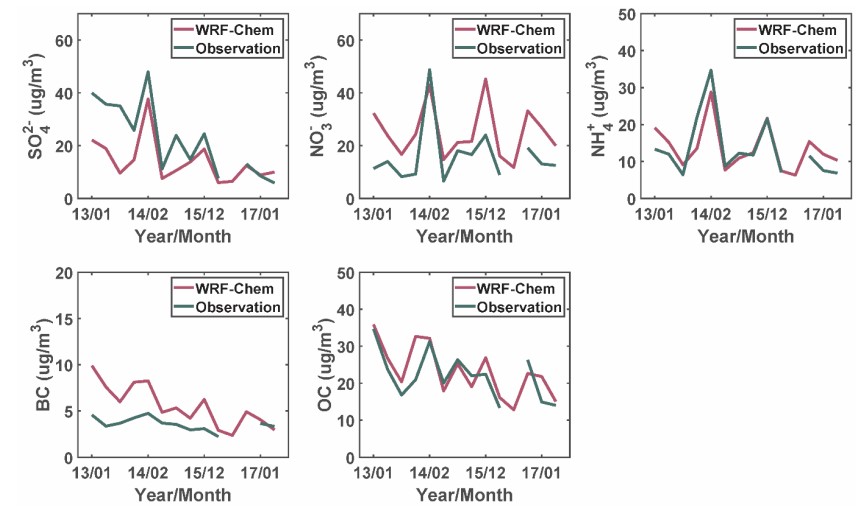

Fig. 3. Simulated and observed temporal variations of monthly mean concentrations of PM$_{2.5}$
chemical from 2013 to 2017.






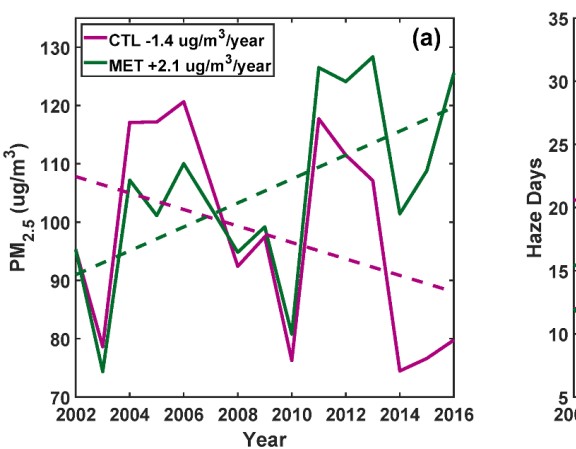
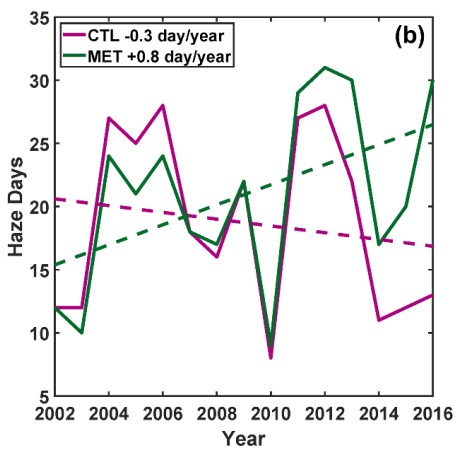

Fig. 4. Wintertime mean concentrations of PM$_{2.5}$ and number of haze days (defined with daily mean concentration above 150 µg/m$^3$) from the CTL and MET simulations.


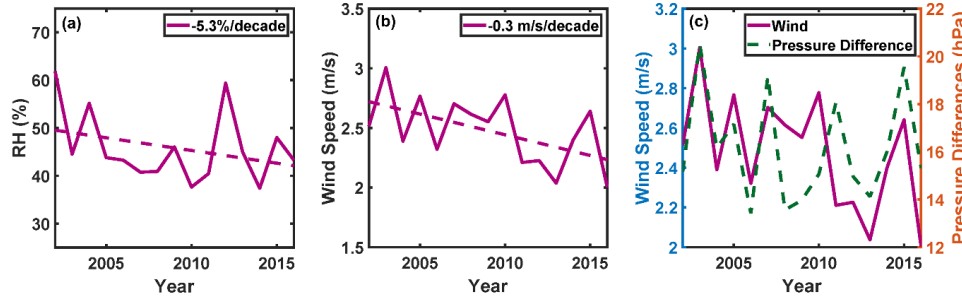

Fig. 5. Simulated winter mean RH, wind speeds in Beijing with declining rates, and pressure difference indicating the intensity of East Asia Winter Monsoon.




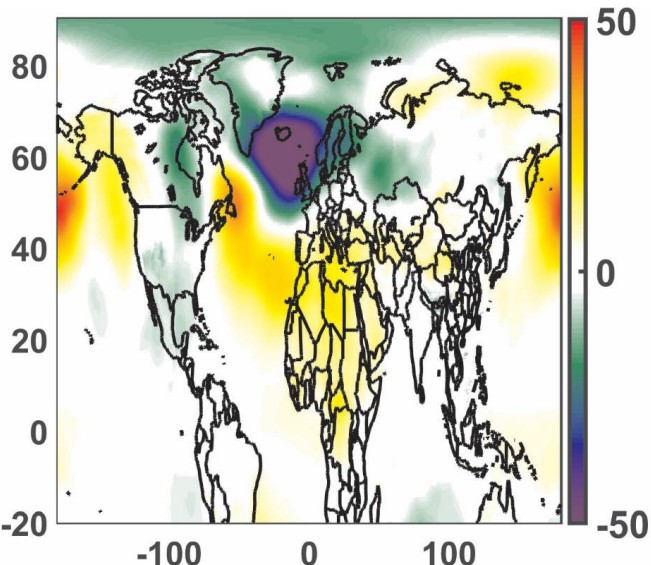

Fig. 6. Trends of winter sea level pressure during 2002-2016 period (unit: Pa/year).




