# Peer review of "China's emission control strategies have suppressed unfavorable influences"

_Atmospheric Chemistry and Physics, 2019_

## Referee Comment (RC1) · Bin Zhu (Referee) · 16 Jun 2019

This is a well-organized and clearly written manuscript on quantified the relative influences of anthropogenic emissions and meteorological conditions on measured PM2.5 concentrations, an important topic in the field of atmospheric environment. It merits to be published after some revisions. 1) Figure 3a showed that the concentrations of PM2.5 have been declining since 2012. However, many studies reported that winter of 2013 was suffered the most serious air pollution in recent years. Is the data used in this study different? 2)In line 220, revise "Over the entire study period" into "Over the entire study period, 2002-2016"; 3) in line 248, what are the reasons of RH decreases

slightly (-5.3 %/decade) between 2002 and 2016 period (Fig. 4a); 4) Is it significant or not that the correlation between wind speed and pressure difference in figure 4c? 5) in the introduction around line 107, I suggest to indicate the increase northly wind could be the potential threaten to air quality over the Yangtze River Delta, China, downwind of north China plain (Kang et. al, 2019). ref. to: Kang, H., Zhu, B., Gao, J., He, Y., Wang, H., Su, J., Pan, C., Zhu, T., and Yu, B.: Potential impacts of cold frontal passage on air quality over the Yangtze River Delta, China, Atmos. Chem. Phys., 19, 3673-3685, https://doi.org/10.5194/acp-19-3673-2019, 2019.

---

## Referee Comment (RC2) · Anonymous Referee #3 · 16 Sep 2019

This manuscript is to understand the effect of changes in meteorology and emission on wintertime PM2.5 concentrations in Beijing. There are some concerns that need to be addressed. 1) There are more journal papers studying effect of meteorology and emissions on air quality in China. The introduction should be more comprehensive. 2) Line 85: Peral -> Pearl. 3) Did the model include dust emissions? If so, it is better to provide the information in the section 2.2. 4) Line 166: the second scenario had varying meteorological conditions and fixed anthropogenic emissions. According to my understanding, by comparing this scenario with the CTL scenario, the difference should reflect the effect of anthropogenic emissions. It is suggested to change the name of the second scenario as "Emis". 5) Section 3.1: Emission inventory, and initial

and boundary meteorological conditions, for different years varied, and thus the model performance for different years also varied. So, the difference of simulated air pollutant levels between years may not fully reflect the trend. In particular, the authors claimed that the PM2.5 in Beijing showed a decreasing trend of 1.4 $\mu$g/m3 per year. This result is within the model error magnitude and uncertainty. 6) The study used 150$\mu$g/m3 as a threshold to define a haze day. However, the model showed an overestimation of PM2.5 that would affect the simulated of haze days. 7) Line2 254-260: the results are not clear. It is claimed that the RH is a good indicator, but it is mentioned that the variability of RH2 is unlikely to be the driver of enhanced PM2.5 under changing conditions of climate. 8) Line 291: It is believed that the effect of meteorology and emissions should not be linear. The study performed two simulations (CTL and MET); however, it is worth to investigate one more simulation with change emissions and fixed meteorology to confirm the results. 9) Table 1: it is surprised that only RH2 and WS10 had a high statistical significance. It is believed BLH should also be one of the major factors that affect air quality. Also, wind direction should play an important role too. 10) Figs. 1 and 6: the aspect ratio of the maps seems to be incorrect.
* * *

---

## Author Comment (AC1) · 29 Oct 2019

**RC1**

This is a well-organized and clearly written manuscript on quantified the relative influences of anthropogenic emissions and meteorological conditions on measured PM$_{2.5}$ concentrations, an important topic in the field of atmospheric environment. It merits to be published after some revisions.

1) Figure 3a showed that the concentrations of PM$_{2.5}$ have been declining since 2012. However, many studies reported that winter of 2013 was suffered the most serious air pollution in recent years. Is the data used in this study different?

Reply: Yes, the winter of 2013 (January 2013) was the most seriously polluted period in recent years. The data used in this study are similar to others. In our definition, winter of 2012 include December of 2012, January of 2013 and February of 2013. We clarified this in lines 157:160, "Wintertime periods defined as the last month of the year and the following two months of the next year were simulated for years 2002-2016. For example, the winter of 2002 includes December of 2002, January of 2003 and February of 2003."

2) In line 220, revise "Over the entire study period" into "Over the entire study period, 2002-2016";

Reply: We have changed it.

3) in line 248, what are the reasons of RH decreases slightly (-5.3 %/decade) between 2002 and 2016 period (Fig. 4a);

Reply: As pointed out by Ding et al. (2014), the decrease of the relative humidity was partly due to the increase of surface temperature. We added this reason in the revised manuscript: "As suggested by Ding et al. (2014), the decrease of RH was partly caused by the increase of surface temperature."

Ding, Y. and Liu, Y.: Analysis of long-term variations of fog and haze in China in recent 50 years and their relations with atmospheric humidity, Science China Earth Sciences, 57(1), pp.36-46, 2014.

4) Is it significant or not that the correlation between wind speed and pressure difference in figure 4c?

Reply: The correlation between WRF simulated wind speeds and pressure difference is 0.49 (p=0.06), and the correlation between observed wind speeds and pressure difference is 0.51 (p=0.05). They are well correlated and p values are around the threshold of significance (0.05). As wind speeds are affected also by other factors including surface roughness, its correlation with pressure differences is not perfect.

5) in the introduction around line 107, I suggest indicating the increase northly wind could be the potential threaten to air quality over the Yangtze River Delta, China, downwind of north China plain (Kang et. al, 2019).

ref. to: Kang, H., Zhu, B., Gao, J., He, Y., Wang, H., Su, J., Pan, C., Zhu, T., and Yu, B.: Potential impacts of cold frontal passage on air quality over the Yangtze River Delta, China, Atmos. Chem. Phys., 19, 3673-3685, https://doi.org/10.5194/acp-19-3673-2019, 2019.

Reply: We added this reference in the introduction in the revised manuscript: "The variability of northly winds in North China has also been linked to air quality over the Yangtze River Delta, which is a downwind region of North China (Kang et al., 2019)".

**RC2**

This manuscript is to understand the effect of changes in meteorology and emission on wintertime PM$_{2.5}$ concentrations in Beijing. There are some concerns that need to be addressed.

1) There are more journal papers studying effect of meteorology and emissions on air quality in China. The introduction should be more comprehensive.

Reply: We added more references with similar topics in the revised manuscript: "The influences of anthropogenic emissions and meteorological conditions on air quality over shorter periods have been investigated extensively (Gao et al., 2017a; Y. Gao et al., 2011; Liu et al., 2017; Wang et al., 2016; Xing et al., 2011)."

Gao, M., Liu, Z., Wang, Y., Lu, X., Ji, D., Wang, L., Li, M., Wang, Z., Zhang, Q. and Carmichael, G.R.: Distinguishing the roles of meteorology, emission control measures, regional transport, and co-benefits of reduced aerosol feedbacks in "APEC Blue", Atmos. Env., https://doi.org/10.1016/j.atmosenv.2017.08.054, 2017.

Gao, Y., Liu, X., Zhao, C. and Zhang, M.: Emission controls versus meteorological conditions in determining aerosol concentrations in Beijing during the 2008 Olympic Games, Atmos. Chem. Phys., 11, 12437-12451, https://doi.org/10.5194/acp-11-12437-2011, 2011.

Liu, T., Gong, S., He, J., Yu, M., Wang, Q., Li, H., Liu, W., Zhang, J., Li, L., Wang, X. and Li, S.: Attributions of meteorological and emission factors to the 2015 winter severe haze pollution episodes in China's Jing-Jin-Ji area, Atmos. Chem. Phys., 17(4), https://doi.org/10.5194/acp-17-2971-2017, 2017.

Wang, Y., Zhang, Y., Schauer, J.J., de Foy, B., Guo, B., and Zhang, Y.: Relative impact of emissions controls and meteorology on air pollution mitigation associated with the Asia-Pacific Economic Cooperation (APEC) conference in Beijing, China, Sci. Total Env., https://doi.org/10.1016/j.scitotenv.2016.06.215, 2016.

Xing, J., Zhang, Y., Wang, S., Liu, X., Cheng, S., Zhang, Q., Chen, Y., Streets, D.G., Jang, C., Hao, J. and Wang, W.: Modeling study on the air quality impacts from emission reductions and atypical meteorological conditions during the 2008 Beijing Olympics, Atmos. Env., https://doi.org/10.1016/j.atmosenv.2011.01.025, 2011.

2) Line 85: Peral -> Pearl.
Reply: Changed.

3) Did the model include dust emissions? If so, it is better to provide the information in the section 2.2.
Reply: Yes, we added in sect. 2.3: "Dust emissions and online sea-salt emissions were also calculated online."

4) Line 166: the second scenario had varying meteorological conditions and fixed anthropogenic emissions. According to my understanding, by comparing this scenario with the CTL scenario, the difference should reflect the effect of anthropogenic emissions. It is suggested to change the name of the second scenario as "Emis".

Reply: In the second scenario, emissions were fixed and only meteorological conditions will affect the simulated concentrations of air pollutants. Thus, we use this simulation to represent the effects of meteorological conditions (MET) on the variations of $PM_{2.5}$. In the presentation of the results, we use the values from CTL and MET simulations directly, instead of the differences between two simulations. We conducted MET simulation to answer the question: if emissions were not changing, what would be the effects of climate conditions on air pollution? To make it clearer, we added clarification in the revised manuscript: "The MET simulation can be used to answer the question: what the climate would have done to the variations of $PM_{2.5}$ if emissions were not changing? The CTL simulation contains the information of changes in both meteorological conditions and emissions."

5) Section 3.1: Emission inventory, and initial and boundary meteorological conditions, for different years varied, and thus the model performance for different years also varied. So, the difference of simulated air pollutant levels between years may not fully reflect the trend. In particular, the authors claimed that the $PM_{2.5}$ in Beijing showed a decreasing trend of 1.4 µg/m3 per year. This result is within the model error magnitude and uncertainty.

Reply: Thanks for pointing out this issue.

Yes, the performance of model varies with years. As seen from Fig. 2, model overestimates concentrations of $PM_{2.5}$ before 2012. Over 2012-2016, model shows reasonably good performance. Over the period 2012-2016, both model and observations show rapid declines. We agree that the decreasing trend of 1.4 µg/m3 might have been overstated given the overestimation of $PM_{2.5}$ in early years.

In the manuscript, we have already acknowledged this issue: "**Given the more serious overestimate of $PM_{2.5}$ in early periods (Fig. 2), the declining rates inferred from the CTL simulation might have been slightly overstated.**" To make it clearer, we added acknowledgement also in the abstract: "Given the overestimation of $PM_{2.5}$ by model, the

effectiveness of stringent emission control measures might have been slightly overstated" in the revised manuscript.

However, this will not change our conclusion that unfavorable climate conditions would have led to more pollution in Beijing, and stringent emission control measures have changed the direction of changes. To make the conclusion more reasonable, we only used statistics drew between 2011 and 2016, during which model shows better performance: "Between the winters of 2011 and 2016, stringent emission control measures resulted in a 21% decrease in mean mass concentrations of PM2.5 in Beijing, with 7 fewer haze days per winter on average. "

6) The study used 150µg/m$^3$ as a threshold to define a haze day. However, the model showed an overestimation of PM2.5 that would affect the simulated of haze days.

Reply: Thanks for pointing out this issue. The overestimation of PM2.5, especially during early years, would lead to overestimated haze days. In the revised manuscript, we also explored what would change if the threshold of 75 µg/m$^3$ was used. As shown in Fig. r1, the trends are similar to the those using 150 µg/m$^3$ as a threshold. We have added this discussion in the revised manuscript: "We explored if different thresholds of haze days would change the findings and found the variations are similar when a threshold of 75 µg/m$^3$ was used (Fig. S2)."

[Figure]

Fig. r1. Wintertime mean concentrations of PM2.5 and number of haze days (defined with daily mean concentration above 75 µg/m$^3$) from the CTL and MET simulations.

It is true that notable overestimation of $PM_{2.5}$ in early years would make the trends of both $PM_{2.5}$ concentrations and haze days different. In the revised manuscript, we added the comparison between simulated and observed number of haze days (Fig. r2). Due to the overestimation of number of haze days by the model, the trend has been overestimated by the model. In fact, the observations show no significant declining trend over 2005-2016. However, this will not change our conclusion that the stringent emission control measures by government has changed the direction of $PM_{2.5}$, especially over 2012-2016 period. We calculated the declining rate of number of haze days inferred from model and observation, and the values are generally consistent (-4.8 day/year and -3.0 day /year). This further indicate that the emission inventory is more accurate over recent years because there were more data to constrain it.

In the revised manuscript, we added the following descriptions to make the presentation clearer: "Due to the overestimation of number of haze days by the model in early years, the declining rates of number of haze days inferred from the CTL simulation might have been overstated. As seen from Fig. S3, there is no notable declining trend in number of haze days inferred from observations over 2005-2016. However, it is consistent that both model and observations indicate rapid declines in number of haze days (-4.8 days and -3.0 days per winter, respectively). With fixed emissions, MET simulation suggests that unfavorable climate conditions would have led to more haze days, emphasizing the significance of emission control in recent years."

[Figure]

Fig. r2. Modeled and observed number of haze days (defined with daily mean concentration above 75 $\mu g/m^3$)

7) Line 254-260: the results are not clear. It is claimed that the RH is a good indicator, but it is mentioned that the variability of RH2 is unlikely to be the driver of enhanced $PM_{2.5}$ under changing conditions of climate.

Reply: RH can explain the variability, but not the increasing trend. RH is positively correlated with $PM_{2.5}$ as higher RH promotes more productions of $PM_{2.5}$. We observed that long-term meteorological conditions have led to higher $PM_{2.5}$, while RH declined over this period. As pointed out by Ding et al. (2014), the decrease of the relative humidity was partly due to the increase of surface temperature. We added this reason in the revised manuscript: "As suggested by Ding et al. (2014), the decrease of RH was partly caused by the increase of surface temperature."

Because the declining trend of RH contradicts with the enhancement of $PM_{2.5}$, we concluded that RH2 is unlikely to be the reason for higher $PM_{2.5}$. However, the declining wind speeds could explain the trend. To make it clearer, we added the following explanation in the revised manuscript. "RH2 has been found to explain the interannual variability of $PM_{2.5}$, but not the increasing trend."

Ding, Y. and Liu, Y.: Analysis of long-term variations of fog and haze in China in recent 50 years and their relations with atmospheric humidity, Science China Earth Sciences, 57(1), pp.36-46, 2014.

8) Line 291: It is believed that the effect of meteorology and emissions should not be linear. The study performed two simulations (CTL and MET); however, it is worth to investigate one more simulation with change emissions and fixed meteorology to confirm the results.

Reply: We agree that the effects of meteorology and emissions are not linear. We performed MET simulation to examine how climate/meteorological conditions will affect air pollution when emissions are not changing. We conducted CTL to explore how variations of emissions would have changed the influence of climate conditions. The purpose of current study is not to investigate how emission changes would change air quality under fixed meteorology as meteorological conditions is difficult to be fixed in reality.

The proposed simulation by the review with fixed meteorology and changing emissions were examined in our previous two studies. In Gao et al. (2016b), we investigated how emission changed over 1960-2010 will affect haze pollution. We found that from 1960 to 2010, the dramatic changes in emissions lead to +260 % increases in sulfate, +320 % increases in nitrate, +300 % increases in ammonium, +160 % increases in BC, and +50 % increases in OC. The responses of $PM_{2.5}$ to individual emission species indicate that the simultaneous increases in $SO_2$, $NH_3$, and $NO_x$ emissions dominated the increases in $PM_{2.5}$ concentrations.

In a recent study just published online (Zhou et al., 2019), we conducted three sets of simulations to explore the changes over 2011/2012-2017/2018.

(1) M11E11: using meteorological data of year 2011/2012, and emission data of year 2011

(2) M17E11: using meteorological data of year 2017/2018 and emission data of year 2011;

(3) M17E17: using meteorological data of year 2017/2018, and emission data of year 2017;

The difference between M17E11 and M17E17 is what proposed by the reviewer. We found that over the period 2011-2018, emission reductions play a more important role than meteorological factors. In the revised manuscript, we added the conclusions in Zhou et al. (2019). "Zhou et al.

(2019) concluded that emission reductions play a more important than meteorological conditions in determining the declines in PM$_{2.5}$ over 2011/2012-2017/2018."

Gao, M., Carmichael, G. R., Saide, P. E., Lu, Z., Yu, M., Streets, D. G. and Wang, Z.: Response of winter fine particulate matter concentrations to emission and meteorology changes in North China, Atmos. Chem. Phys., 16, 11837-11851, https://doi.org/10.5194/acp-16-11837-2016, 2016b.

Zhou, W., Gao, M., He, Y., Wang, Q., Xie, C., Xu, W., Zhao, J., Du, W., Qiu, Y., Lei, L., Fu, P., Wang, Z., Worsnop, D.R., Zhang, Q. and Sun, Y.: Response of aerosol chemistry to clean air action in Beijing, China: Insights from two-year ACSM measurements and model simulations, Env. Pol., doi: https://doi.org/10.1016/j.envpol.2019.113345, 2019.

9) Table 1: it is surprised that only RH2 and WS10 had a high statistical significance. It is believed BLH should also be one of the major factors that affect air quality. Also, wind direction should play an important role too.

Reply: I agree that BLH is an important factor that affects the occurrence of air pollution episodes. As we showed in previous study (Gao et al., 2016, Fig. 9d) that the concentrations of PM$_{2.5}$ have an inverse relationship with BLH. In this study, we are looking at long-term trend, and only wintertime mean concentrations of PM$_{2.5}$ and meteorological factors are considered instead of hourly mean or daily mean. We found that long-term BLH is not as important as WS10 and RH2 in determining the long-term trend and interannual variability of PM$_{2.5}$.

Cai et al. (2017) also examined the correlation between BLH and long-term PM$_{2.5}$ and found they are not highly correlated.

Yes, wind direction is an important factor. As mentioned in our previous study (Gao et al., 2016) that haze episodes happen under continuous southerly winds and end under strong northerly winds. The wind directions affect air quality through two ways: (1) in wintertime, speeds of southerly winds are weaker than speeds of northerly winds; (2) southerly winds can bring humid air from the ocean, which can accelerate secondary aerosol formation. As shown in this study that pollution has a close connection with RH. Thus, the influences of wind direction have been contained in the influences of wind speeds and RH.

Gao, M., Carmichael, G.R., Wang, Y., Saide, P.E., Yu, M., Xin, J., Liu, Z. and Wang, Z., 2016. Modeling study of the 2010 regional haze event in the North China Plain. Atmospheric Chemistry and Physics, 16(3), p.1673.

Cai, W., Li, K., Liao, H., Wang, H. and Wu, L., 2017. Weather conditions conducive to Beijing severe haze more frequent under climate change. Nature Climate Change, 7(4), p.257.

10) Figs. 1 and 6: the aspect ratio of the maps seems to be incorrect.

Reply: We've updated Figs. 1 and 6 in the revised manuscript.

[Figure]

Fig.1. WRF-Chem modeling domain settings and locations of observations.

[Figure]

Fig. 6. Trends of winter sea level pressure during 2002-2016 period (unit: Pa/year).

---

## Author Response (AR2)

The revised manuscript and response letter have fairly addressed the reviewers' comments. Now following the reviewer 2's discussion, I have one more concern that I hope the authors can address to improve the manuscript. The key message of this study is as claimed by the title, "China's Clean Air Action has suppressed unfavorable influences of climate on wintertime $PM_{2.5}$ concentrations in Beijing since 2002". The Clean Air Action were enforced during the period of 2013-2017, while it was used by the authors to explain the trends of $PM_{2.5}$ levels and haze days in Beijing over winters 2002-2016. The inconsistency between the two time periods needs caution. Based on the two sensitivity experiments (CTL: varying emissions, and MET: 2002 fixed emissions), the study did only separate the role of varying anthropogenic emissions over the whole period of 2002-2016, and did not separate the role of emission reductions due to the Clean Air Action alone over 2013-2017. Thus, attributing the trends over 2002-2016 to the Clean Air Action may not fully correct based the current model experiments. One evidence to support this argument is in Fig 4, the turning points driven the trend of PM2.5 were in 2010 and 2011, these were before the Clean Air Action. Therefore, I think what suppressed the unfavorable climate influences is not the Clean Air Action alone, but all emission control strategies over 2002-2016, based on the analyses in the present manuscript.

Response: Thanks for careful reading and mentioning this very important issue. We agree that the Clean Air Action might not be a good description in this context.
However, emission control measures started even before 2013. Before 2013, strengthening the emission standards for power and industrial sectors was the key pollution control measure (Zheng et al., 2018).
In the revised manuscript, we have replace "clean air action" with "Emission control strategies" to avoid confusion.

Zheng, B., Tong, D., Li, M., Liu, F., Hong, C., Geng, G., Li, H., Li, X., Peng, L., Qi, J., Yan, L., Zhang, Y., Zhao, H., Zheng, Y., He, K., and Zhang, Q.: Trends in China's anthropogenic emissions since 2010 as the consequence of clean air actions, Atmos. Chem. Phys., 18, 14095–14111, https://doi.org/10.5194/acp-18-14095-2018, 2018.

Other Comments:

Page 8, Line 241: The two sentences here "there is no notable declining trend in number of haze days inferred from observations over 2005-2016" and "it is consistent that both model and observations indicate rapid declines in number of haze days (-4.8 days and -3.0 days per winter, respectively)" are inconsistent. Was the second sentence for the period of 2011-2016?
Response: Yes, the second sentence was for the period of 2012-2016. We have added it to the second sentence.

Page 9, Line 275:
What is the difference of RH and RH2? Please clarify.

Response: RH denotes relative humidity in general, while RH2 represents relative humidity at 2 meters, which was analyzed in this study. In the revised manuscript, we have added clarification about RH, but replace RH with RH2 in most sentences to avoid confusion.

Page 10, Figure 5:
What is a blue y-axis "wind speed" in Fig 5 panel (c)for? Is it the same as panel (b) and why not merge panel (b) and panel (c)?

Response: Yes, it is the same as panel (b). Following your suggestion, we have merged (b) and (c) in the revised manuscript.

[revised manuscript text omitted]